# Influencing Behavioral Attributions
# to Robot Motion During Task Execution

**Nick Walker, Christoforos Mavrogiannis, Siddhartha Srinivasa, Maya Cakmak**
Paul G. Allen School of Computer Science & Engineering, University of Washington, Seattle, USA
`{nswalker,cmavro,siddh,mcakmak}@cs.uw.edu`

**Abstract:** While prior work has shown how to autonomously generate motion that communicates task-related attributes, like intent or capability, we know less about how to automatically generate motion that communicates higher-level behavioral attributes such as curiosity or competence. We propose a framework that addresses the challenges of modeling human attributions to robot motion, generating trajectories that elicit attributions, and selecting trajectories that balance attribution and task completion. The insight underpinning our approach is that attributions can be ascribed to features of the motion that don't severely impact task performance, and that these features form a convenient basis both for predicting and generating communicative motion. We illustrate the framework in a coverage task resembling household vacuum cleaning. Through a virtual interface, we collect a dataset of human attributions to robot trajectories during task execution and learn a probabilistic model that maps trajectories to attributions. We then incorporate this model into a trajectory generation mechanism that balances between task completion and communication of a desired behavioral attribute. Through an online user study on a different household layout, we find that our prediction model accurately captures human attribution for coverage tasks.

## 1 Introduction

As robots enter households and public spaces, it is increasingly important to account for human perceptions of their behavior [1, 2, 3, 4]. While a robot's actions might be driven by unambiguous internal objectives, solely optimizing such criteria might result in behavior that is difficult to interpret or disruptive to human onlookers. For example, a highly articulated robot may follow a non-humanlike trajectory that users attribute caprice to, making observers uncomfortable [5], or a home robot such as a robot vacuum cleaner can turn arbitrarily and cause observers to perceive the robot as broken, interrupting home activity.

Accounting for high-level attributions to robot behavior is a complex problem relying on the mechanisms underlying human attribution and behavior generation. In psychology, there is a long history of work on understanding human attribution for behavior explanation or inference of behavior traits [6, 7]. Humans are highly attuned to how their actions are perceived and adapt their behavior to elicit a desired impression from others or adhere to social norms, a concept that is known as *presentation of self* [8]. The tendency for humans to attribute even situational behaviors to deeper character traits is so pervasive that it is known as "the fundamental attribution error" [9].

Inspired by these theories, we envision robots that can 1) leverage an understanding of humans' attribution mechanisms to predict attributions to their motion, 2) generate behaviors that elicit desired human impressions and 3) balance attribution elicitation and task completion. This will increase the acceptance of robots in human spaces by enabling them go about their tasking in a way that is both effective and sensitive to perceptions.

We propose a framework, shown in Fig. 1, that addresses these challenges by integrating a learned model of human attribution into a robot's trajectory generation process. Our observation is that users rely on local characteristics of the robot's motion, like short patterns of actions, in combination with global trajectory characteristics, like the amount of redundancy, to infer attributions. These features distinguish otherwise functionally equivalent trajectories and provide a basis for the framework. We

5th Conference on Robot Learning (CoRL 2021), London, UK.

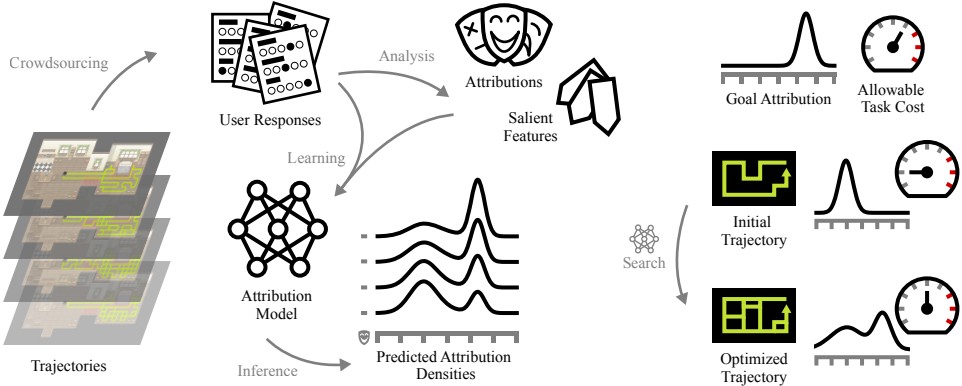

Figure 1: Our proposed framework. User responses to robot trajectories are analyzed to extract salient features and attributions, then used to train a model that probabilistically maps robot trajectories to human attributions (left). The acquired model is used to generate robot trajectories that elicit a desired attribution (right).

trace an application of the framework to a virtual robot vacuum cleaning task. In our evaluation, we see that the resulting model is useful for predicting attributions and for enabling the generation of trajectories that balance task execution and attribution elicitation. Some aspects, such as partial trajectory observation or application to highly articulated robots, remain for future work.

## 2 A Framework for Behavioral Attribution

We consider a robot performing a task $G$ in a human environment. We denote by $s \in \mathcal{S}$ the robot state where $\mathcal{S}$ is a state space, and define a robot trajectory as a sequence of states $\xi = (s_0, \ldots, s_t)$ where indices correspond to timesteps following a fixed time parametrization. Let us define the task as a tuple $G = (\Xi, \mathcal{A}, \mathcal{P}, C)$ where $\Xi$ is a space of robot trajectories, $\mathcal{A}$ denotes the robot action space, $\mathcal{P} : \Xi \times \mathcal{A} \to \Xi$ represents a deterministic state transition model, and $C : \Xi \to \mathbb{R}$ is a trajectory cost. We assume that the robot starts from an initial state $s_0$ and reaches a terminal state $s_T$ (at time $T$) by executing a trajectory $\xi = (s_0, \ldots, s_T)$. We assume that this trajectory $\xi$ is fully observed by a human who is aware of the task specification $G$.

The observer makes an inference of the form $\mathcal{I}_B : \Xi \times \mathcal{G} \to \mathcal{B}$, mapping their observation from the space of trajectories $\Xi$, along with the context of the task specification $G \in \mathcal{G}$, into a space of behavioral attributions $\mathcal{B}$. The form of $\mathcal{B}$ will vary, but should be selected to capture the range, combinations, and intensities of attributions that the robot should be sensitive to.

Conversely, we can imagine that a robot, given a behavioral attribution $b \in \mathcal{B}$ and a task $G$, infers a trajectory $\xi_b \in \Xi$ that exemplifies the attribution, corresponding to an inference of the form $\mathcal{I}_\xi : \mathcal{B} \times \mathcal{G} \to \Xi$. In other words, we assume that there is a "way" that a curious—or any other attribution—robot should execute a particular task. In practice, we will realize both of these inferences as probabilistic maps. Rather than solely capturing the best way to look curious for a task, we'll seek to assign densities to trajectories, allowing the possibility that there are many equally likely alternatives.

In the remainder of paper, we aim to provide a general framework for modeling inferences of the form $\mathcal{I}_B$, and $\mathcal{I}_\xi$. Our goal is to enable robots to understand and account for the communicative effects of their motion on human observers.

## 3 Related Work

Several studies have illustrated the value of robot motion as a communicative modality [10, 11, 12, 13, 14, 15]. Some works propose algorithms for legible robot motion generation, which have been shown to enable effective human-robot collaboration in manipulation tasks [10], or smooth robot navigation in close proximity to humans [12, 14]. Other works focus on conveying higher-level information such as the robot's objective function [15] or the source of failure [13] when the robot

can't complete a task. Animation principles [16] or movement analysis [17] are often employed to inform the design of expressive robot behaviors. Finally, related graphics research focuses on the generation of stylistically distinct but functionally equivalent motion primitives for walking and other activities [18, 19].

The complex interplay of embodiment and communicative motion has motivated research on understanding human perceptions of robot behavior. For instance, early work looked at the effect of robot gaze on human impressions [20]. Sung et al. [1] study human attitudes towards robot vacuum cleaners and propose design principles aimed at enhancing the acceptance of robots in domestic environments. [2] report a relation between robot motion and perceived affect. Lo et al. [21] and [14] investigate human perceptions of different robot navigation strategies whereas Walker et al. [22] study human perceptions of robot actions that deviate from the robot's assigned task.

Our work draws inspiration from previous efforts to characterize human perceptions and attributions to robot motion [1, 2]. However, it moves beyond the problem of understanding and analyzing human perceptions and focuses on the problem of *synthesizing* implicitly communicative motion. Our work is closely related to past work on the generation of legible robot motion [10, 11, 23, 13] in that we also incorporate a model of human inference into the robot's motion generation pipeline. However, unlike these works which emphasize the communication of task-related attributes, our focus is instead on communicating high-level, behavioral attributes through robot motion.

## 4 Modeling and Influencing Attributions

We consider a scenario in which a mobile robot performs a coverage task in a two-dimensional discrete workspace while a human is observing from a top-down view. We employ a virtual environment[1] that resembles a house and stylize the agent as a robot vacuum cleaner (see Fig. 2) since the general population is already somewhat familiar[2] with such robots [1, 24], making it easier for participants to develop mental models about their motion than that of a manipulator, for example.

In this scenario, the robot state space is the complete home workspace and $\Xi$ is the space of all possible trajectories of any length that could be followed in the space. The robot action space $\mathcal{A}$ consists of deterministic movements in the cardinal directions. The cost of a state transition from a state $s_t$ to a state $s_{t+1}$ after having followed a trajectory $\xi_t$ is defined as 0 if the state hasn't been visited before, -5 if the state contains a small, traversable obstacle (e.g. a vase), and -1 otherwise. A penalty of 3 times the number of unvisited states from the goal region is applied on termination.

### 4.1 Understanding Behavioral Attribution for Coverage Tasks

Through exploratory studies on Amazon Mechanical Turk, we sought to extract domain knowledge for attributions to robot motion within coverage tasks. Using the home layout shown in Fig. 2, we generated a set of trajectories exhibiting qualitatively distinct ways the robot could respond to the prompt to "clean the bedroom," ranging from a near optimal coverage plan to a trajectory that barely visited the target room (see Appendix A for additional details). Each participant viewed videos of a random selection of three of these trajectories. After each video, participants were asked: a) to provide three words to describe the robot's behavior; b) to rate their agreement that "the robot is ______" for a range of adjectives drawn from relevant literature on human attributions [25, 26, 22]; c) to "explain what factors contributed to their strongest ratings." In addition to attributions, participants were asked to use an interactive interface to demonstrate how they would "clean the bedroom in a way that makes the robot look ______" where the blank was filled with a random adjective from the attribution rating items. Across all exploratory studies, we collected 375 sets of attribution ratings from 115 participants (73 male, 41 female) aged 21-70 ($M$ = 38.3, $SD$ = 10.7) covering 63 trajectories and a total of 193 demonstrations.

#### 4.1.1 Extracting the Space of Attributions

To understand the inter-correlation of participant adjective ratings, we conducted an exploratory factor analysis. We selected a three-factor, promax rotation model which explained 74% of the

---

[1]The environment is built in the Phaser game engine (https://phaser.io/) and uses art by Bonsaiheldin under a CC-BY-SA license. It and the rest of paper's code can be found in the supplement.

[2]A presentation by iRobot [24] estimates that ~19M U.S. households had robot vacuum cleaners in 2020.

| Feature | Description |
|---|---|
| **Coverage** (%) | Goal region states visited at least once. |
| **Redundant coverage** (%) | Goal region states visited more than once. |
| **Overlap** (%) | Plan states visited more than once. |
| **Length** (%) | Normalized plan length. |
| **Hook template** (%) | Frequency of "U" shape patterns in plan. |
| **Straight template** (%) | Frequency of action repetition in plan. |
| **Start-stop template** (%) | Frequency of idle-move-idle patterns in plan. |
| **Idleness** (%) | Frequency of idle actions in plan. |
| **Map coverage** (%) | Fraction of map states visited at least once. |
| **Collision** (%) | Fraction of obstacle states from $O$ in plan. |
| **Goal deviation** (%) | Fraction of plan before first goal state. |

Table 1: Low-dimensional trajectory representation.

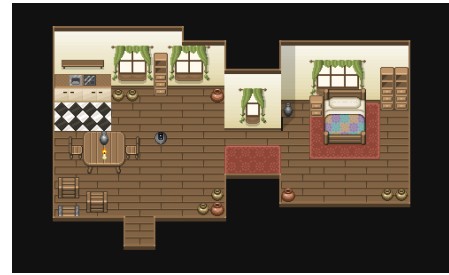

Figure 2: The home environment used in our exploratory studies.

observed variance due to its parsimony and coherence (see Appendix B). The first factor, which we call "competence" for its similarity to the relevant factor described by Carpinella et al. [26] consists of six items (responsible, competent, efficient, reliable, intelligent, focused) centered on the capability and diligence of the robot. The second consists of four items (lost, clumsy, confused, broken) alluding to a negative state, for which we title the factor "brokeness". The third contains two items (curious, investigative) and matches the curiosity factor examined by Walker et al. [22].

The extracted model enables the computation of standardized factor scores roughly in the range $[-3, 3]$ which denote how many standard deviations from the mean a participant's ratings for the items are. Reflecting the format of the component items, a high or low factor score denotes agreement or disagreement that a trajectory expresses an attribution, respectively. Based on this model, we represent the attribution for a trajectory $\xi$ as a tuple $\boldsymbol{b} = (b_{competent}, b_{broken}, b_{curious}) \in \mathcal{B}$ where the space of attributions is the set $\mathcal{B} = [-3, 3]^3$.

### 4.1.2 Low-dimensional Trajectory Representation

The space of possible trajectories in this domain is too large to map directly to the space of attributions, so we constructed a low-dimensional space $\Phi$ based on features relevant to the formation of attribution ratings. This allows us to describe a trajectory $\xi$ as a vector $\phi_\xi = \phi(\xi) \in \Phi$. The feature space was inspired by relevant literature on human behavioral attribution to robot motion and enriched with features appearing in participants' explanations. The final set of 11 features used in further experiments is listed in Table 1.

### 4.2 Mapping Trajectories to Attribution Scores

Given a trajectory $\xi$, an observer's inference $\mathcal{I}_B$ of behavioral attribution can be expected to vary both due to individual differences and as a result of measurement error. For this reason, we model $\mathcal{I}_B$ as a conditional probability density $f_{\mathcal{B}|\Xi}(\boldsymbol{b}|\phi_\xi) : \mathcal{B} \to \mathbb{R}$. We observed multimodality in the distribution of factor scores for some trajectories, so we use a Mixture Density Network (MDN) [27] to approximate each conditional density as a mixture distribution $f_{\mathcal{B}|\Xi}(\boldsymbol{b}|\phi_\xi) = \sum_{i=1}^{C} \alpha_i(\phi_\xi) k_i(\boldsymbol{b}|\phi_\xi)$ where $\alpha_i$, $i = 1, \dots, C$, is a mixing coefficient, and $k_i$ is a multivariate Gaussian kernel function with mean $\boldsymbol{\mu}_i$ and covariance $\Sigma_i$. Note that the mixing coefficients $\alpha_i$ and the Gaussian parameters $\boldsymbol{\mu}_i$ and $\Sigma_i$ are functions of the featurized trajectory $\phi_\xi$. In our models, these functions are implemented as linear transformations of features produced by a shared multi-layer perceptron.

To make efficient use of scarce data, we created ensembles of MDNs using bootstrap aggregation, i.e., we trained $N$ models with different data splits and uniformly weight their predictions: $f_{\mathcal{B}|\Xi}^{ens}(\boldsymbol{b}|\phi_\xi) = \frac{1}{N} \sum_{i=1}^{N} f_{\mathcal{B}|\Xi}^{i}(\boldsymbol{b}|\phi_\xi)$.

We studied three different model configurations; single and four component MDNs, i.e., $C = 1$ and $C = 4$ and an ensemble of 8 MDNs each with four components, i.e., $C = 4$, $N = 8$. We trained all models using an average negative log likelihood (NLL) loss function, the Adam optimizer [28], noise regularization [29], and early stopping. We configured the input MLP to use a single hidden layer with 5 units and a hyperbolic tangent activation. We expanded the dataset collected in our exploratory studies after assessing the sensitivity of the models to increased amounts of data, a process described in Appendix C. The final version of the set includes 126 trajectories with 671

| Model | Parameters | Average NLL | SD |
|---|---|---|---|
| Uniform | 6 | 5.38 | 0.00 |
| MDN, C=1 | 120 | $3.13 \pm .05$ | $1.35 \pm .09$ |
| MDN, C=4 | 300 | $2.66 \pm .08$ | $1.57 \pm .05$ |
| MDN Ensemble, C=4 N=8 | 2400 | $2.53 \pm .06$ | $1.38 \pm .04$ |

Table 2: Average test negative log likelihood (NLL) for each model configuration. Each datapoint represents a mean NLL over 16 models trained with random train-validate folds on a fixed test set. Error is the 95% confidence interval calculated with bootstrapping.

attribution ratings. Table 2 compares the NLL of the models over held-out data. The mean indicates the typical quality of the prediction and the standard deviation indicates the degree to which this varied from sample to sample. Both quantities are averaged over 16 random folds and reported with bootstrapped 95% confidence intervals. All models perform significantly better than a uniform baseline, which simply assigns equal probability to all outcomes. The ensemble model performs best and is used in further experiments in the remainder of the paper.

### 4.3 Generating Trajectories that Elicit Desired Attributions

We represent the behavior specification as a one-dimensional Gaussian $b^* \sim \mathcal{N}(\mu_b, \sigma_b^2)$ centered on a desired rating $\mu_b \in [-3, 3]$ for a single attribution dimension where the variance $\sigma_b$ serves as a tolerance parameter modeling the acceptable distance from the desired behavioral rating. We use a density representation as it more closely matches the output of our model for $\mathcal{I}_B$.

Together with the task requirements as described by the cost function $\mathcal{C}$, we realize the inference $\mathcal{I}_\xi$ as an optimization of the form:

$$\begin{aligned} \xi^* = \arg\min_{\xi \in \Xi} \quad & D_{\mathrm{KL}}(f_{\boldsymbol{B}_i} || \mathcal{N}(\mu_b, \sigma_b^2)) \\ \text{s.t.} \quad & \mathcal{C}(\xi) \le w, \end{aligned} \tag{1}$$

where $D_{\mathrm{KL}}$ denotes the KL divergence, $f_{\boldsymbol{B}_i}$ is the density $f_{\boldsymbol{B}|\Xi}|\phi(\xi)$ marginalized across dimensions other than $i$, and $w$ is the maximum allowable task cost. Because many applications are conventionally exclusively task-cost driven, this format provides an intuitive "knob" in the form of how suboptimal the robot is allowed to be. Where performance is critical—perhaps to meet a schedule or to fit in power constraints—the robot designer need only set $w$ to express the hard bound. Different attributions are expected to be more or less sensitive to the allowable suboptimality, something illustrated in Appendix D.

We implement this optimization using a hill-descending search in the space of trajectories. The search is initialized with a task-optimal trajectory generated via A* search and progressively samples modifications to the trajectory. These modifications consist of both naive, action-level modifications to the trajectory as well as changes targeting the activation of the features underlying the attribution model (see Fig. 1). Important motion templates, like runs of straight motion, hook patterns, and start-stops are sampled and patched into trajectories. All modifications are ranked by the divergence of their predicted attribution with the behavior specification. The search terminates after a fixed duration and the best performing trajectory subject to the task-cost constraint is returned. A detailed description of the optimization procedure is given in Appendix D.

## 5 Evaluation

We conduct a user study to evaluate the efficacy of the framework as a means of producing trajectories that elicit desired attributions. Our study is motivated by the following hypotheses:

**Hypothesis 1** The model makes accurate predictions about the distribution of attributions to new trajectories.

**Hypothesis 2** The model makes accurate predictions about the distribution of attributions to trajectories in unseen environments.

**Hypothesis 3** The approach enables the generation of trajectories that elicit desired attributions.

## 5.1 Experiment Design

In Experiment I, participants observe and rate trajectories in the same home layout used for data collection, while in Experiment II, trajectories are generated in a modified home layout. In an effort to understand the impact of the environment geometry, the modified layout increases the size of the goal region by 100%, varies the placement of items and obstacles, and flips the dominant direction of the robot's motion. The experiments are within-subjects, video-based user studies, both instantiated in three parallel sets corresponding to the three attribution dimensions considered. For each dimension, we consider four robot trajectories generated by optimizing (1) with a target distribution expressed as a Gaussian centered at 1.5 with scale 0.3 and varying task cost thresholds. To ease interpretation, the $w$ values governing the thresholds were set in multiples—**1x**, **2x**, **4x**, **12x**—of the cost of the optimal trajectory for the task. The full set of trajectories is shown in Fig. 3[3]. In all experiments, participants rate and describe each trajectory using the same items and questions used in the exploratory studies of Sec. 4. After watching all trajectories in a randomly assigned order, they also respond to additional comparative questions: "which robot seemed the most ______" and "which robot seemed the least ______", where the blanks are filled with the adjective corresponding to the dimension of attribution studied. Both comparisons are accompanied with an open-ended question asking for a brief explanation of the choice. No suitable baselines exist for the balanced attribution elicitation task, so our experiments use solely trajectories generated by our approach.

**Participants** A total of 144 participants (76 male, 68 female) aged 20-72 ($M$ = 38.2, $SD$ = 10.9) were recruited via Amazon Mechanical Turk and paid $2 to complete the approximately 15 minute task. 9 had taken part in our earlier exploratory studies. Participants were equally distributed amongst the six sets of conditions. Condition orderings were fully counterbalanced.

## 5.2 Results

The predictive performance of the model is illustrated in Fig. 5 and reported in Table 3. The 95% bootstrapped confidence interval on the mean NLL for Experiment I was 2.77±.10 ($SD$ = 0.88±.09). and 2.87±0.10 ($SD$ = 0.86±.09) for Experiment II. Participants' choices for "most" and "least" trajectories are shown in Fig. 4.

**Hypothesis 1** was supported; the average NLL of the models was significantly lower than a uniform model, indicating that the model was able to meaningfully predict attributions in the layout it was trained in.

**Hypothesis 2** was supported; the average NLL of the attributions observed across Experiment II was significantly lower than the uniform model, indicating that the model remains informative even under modifications to the environment layout.[4]

**Hypothesis 3** saw mixed support; Kendall's tau-b correlation tests (reported in Appendix E) indicate strong positive correlations between the allowable suboptimality and brokenness factor scores, suggesting that the trajectory generation method was effective at eliciting progressively higher factor scores. However, tests for the competence conditions indicated a moderate negative correlation, and tests for curiosity conditions were not significant. As shown in Fig. 4, while participants found that 12x and 1x were the most and least "______" for experiments focused on curiosity and brokenness, this relationship was flipped for the competence experiments.

When optimizing for competence, the model emphasizes over-coverage of the goal region as well as coverage of the house as a whole (see Fig. 3). Due to the associated task-cost penalty, coverage outside of the goal begins to appear in the 4x condition of Experiment I and the 12x condition of Experiment II, and participants' responses indicate that it is a key driver of negative attributions of competence. Some emphasized that time spent not cleaning the bedroom was "wasted movement in the wrong room" (Exp. I-Competence), while others attributed the deviation to being "lost" or "totally confused" (Exp. I-Competence). In less extreme conditions, users expressed uncertainty about what drove the behavior, saying they were "not sure if robot is cleaning outside the bedroom cause

---

[3]Videos of the trajectories are included in the supplement.

[4]See Appendix E for supplementary tests indicating insufficient evidence to support a significant difference between the predicted and observed distributions in both experiments.

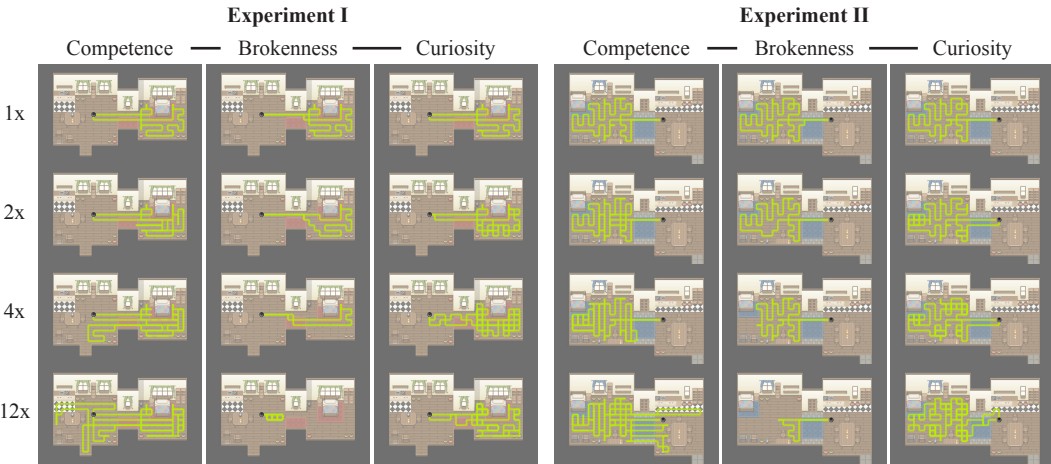

Figure 3: Traces of robot trajectories used in different experiments and conditions.

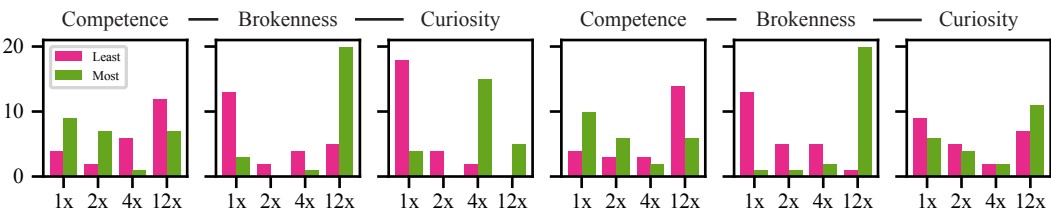

Figure 4: Counts of trajectories picked as most and least "______".

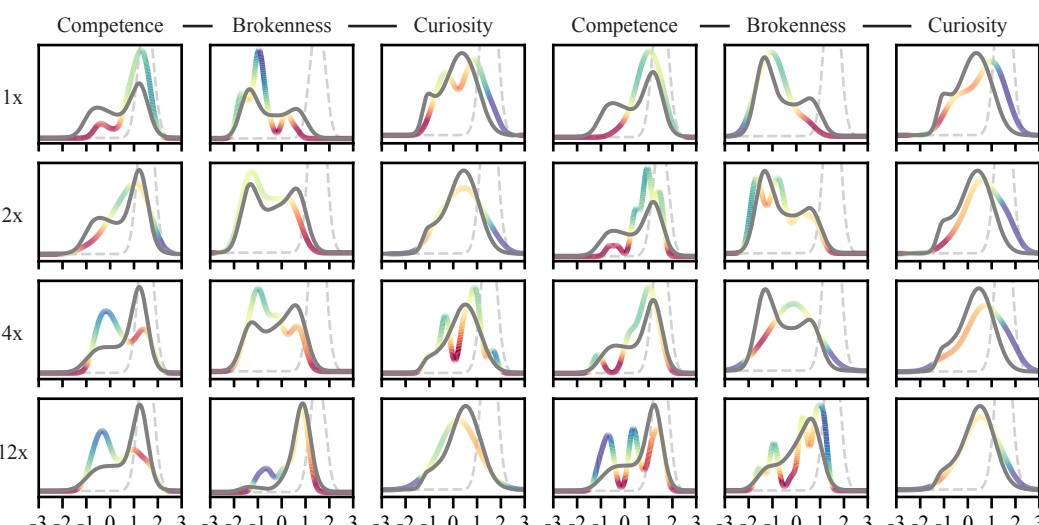

Figure 5: Comparisons of the model's predicted density for the factor score under consideration (in grey), the observed distribution (multicolored) approximated with Gaussian kernel density estimation and the target distribution (dashed). Each subplot has a unique y-scale, with the magnitude of the difference between the predicted and observed densities at any point encoded instead in the color of the line for the observed distribution. Deep red indicates the model severely overpredicted the density, while deep blue indicates severe underprediction.

|       | Competence  | Brokenness  | Curiosity   | Competence  | Brokenness  | Curiosity   |
|-------|-------------|-------------|-------------|-------------|-------------|-------------|
| 1x    | 2.51 (0.69) | 2.44 (0.76) | 2.37 (0.54) | 2.73 (0.59) | 2.64 (0.73) | 2.76 (1.05) |
| 2x    | 2.76 (0.85) | 2.84 (0.72) | 2.85 (0.75) | 2.95 (0.70) | 2.86 (0.88) | 2.85 (1.09) |
| 4x    | 2.98 (0.76) | 2.93 (0.83) | 3.02 (0.71) | 3.00 (0.67) | 2.92 (0.42) | 3.10 (0.98) |
| 12x   | 3.10 (0.73) | 2.53 (1.75) | 2.94 (0.72) | 3.43 (0.84) | 2.28 (0.98) | 2.84 (0.84) |

Table 3: Evaluation Average NLL (SD)

there might be dirt that can be brought in, or just confused as to parameter of bedroom" (Exp. I-Competence, 4x). A minority of users thought that the extra motion was worthwhile, marking the 12x trajectory as the most competent because it "cleaned more areas in both rooms" or "completed the entire home from bedroom to kitchen" (Exp. I-Competence). The model overestimated the prevalence of people that would appreciate the additional coverage of the environment, as indicated by both the "most/least" selections and the NLL. We speculate that the format of the experiment—wherein participants view the optimal trajectory and mentally anchor their ratings against it—is a contributing factor; participants may be more inclined to rate the robot as competent when viewing the optimized trajectories in real-world settings where direct comparison to the task-optimal trajectory is less likely.

Trajectories optimized to look broken progressively cover less of the goal region before ultimately devolving into repeated circular motion near the start point (see Fig. 3). The majority of users concurred in their assessment of the 12x condition as "defective", with some remarking that it seemed overwrought, "a joke version of the robot" (Exp. II-Brokenness). The model underpredicted the extent of participants' agreement that the optimal trajectory would be perceived as not broken, but the predictive performance across all brokenness conditions was still the strongest of the three factors.

When optimizing for curiosity, the model emphasized over-coverage of the goal region, overlapping motion, hook-like patterns and visiting penalized states depicted with vases (see Fig. 3). Some participants highlight the extra coverage as the reason for selecting the 12x as the most curious condition, saying the robot "cleans very well but cleaned the same place multiple time, roaming without reason" or that it "dawdled around a lot, getting hyper fixated on certain spots" (Exp. II-Curiosity). The same factor is highlighted by participants in less extreme conditions, with one speculating of the 4x trajectory that "maybe something caught its eye while it was working and it got so distracted that he totally kept getting off track" (Exp. I-Curiosity). The change in the distribution of curiosity factor scores was expected to be small and the observations bear the predictions out, though it is notable that despite the subtle differences a majority of participants select either 4x or 12x as the "most curious" trajectory across both experiments.

## 6   Discussion

Our framework is a general approach for endowing robots with a sensitivity to the behavioral attributions their motion elicits. We illustrated its application to coverage motions, but we envision a lively stable of related instances stretching across domains from delivery service robots to robot arms in fulfillment centers. The process is the same; the robot arm's designer will build a pool of videos and study users' responses to—and their reasoning about—the motion over a broad set of dimensions, then learn a forward mapping from the features driving their reasoning to the attributions in their responses. While some features such as path length or redundancy may map over from the coverage domain, others like the shape of the acceleration profile may need to be added to capture perceptions of danger or erraticism. The reverse translation from a desired attribution to a new trajectory can be realized by searching in the space of trajectories, using the attribution features to guide the process—something that may have a pronounced impact in a higher dimensional planning space.

We haven't yet addressed some important aspects of behavioral attribution. While our results showed that the approach's performance transferred to a similar environment, future work should use more disparate environments to determine the limits of its generalization. In the setting we explored, the observer looks at the robot's motion from a top-down perspective, but different perspectives may result in different impressions. Further research should evaluate data collection techniques and environments that account for variability in observer perspectives. The features and the learned mappings from features to attributions are specific to the types of environments and the task considered and would likely need to be augmented to work more broadly. We imagine that, in the future, robot designers will have access to a wide array of well-studied features with which to bootstrap their system, and when human-robot interaction data is abundant, we may see the rise of learned representations that can power the understanding and generation of motion with minimal additional supervision.

## Acknowledgments

This work was (partially) funded by the National Science Foundation IIS #2007011, National Science Foundation IIS #1924435 "Program Verification and Synthesis for Collaborative Robots," National Science Foundation DMS #1839371, the Office of Naval Research, US Army Research Laboratory CCDC, Amazon, and Honda Research Institute USA. This material is based upon work supported by the National Science Foundation Graduate Research Fellowship under Grant #1762114. We thank the scientific Python Community for developing the core set of tools that enabled this work, including NumPy [30], SciPy [31], Pingouin [32], PyTorch [33] and Matplotlib [34].

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
