# OpenReview forum: "Influencing Behavioral Attributions to Robot Motion During Task Execution"
_robot-learning.org/CoRL/2021/Conference — CoRL2021 Poster_

### Official Review · Reviewer_zJU5 · 2021-07-07

**Originality:** Good
**Technical Quality:** Good
**Clarity Of Presentation:** Good
**Impact:** 2

**Recommendation:**

Weak Accept: I recommend accepting the paper, but will not argue for my recommendation if the majority of other reviewers have a different opinion.

**Summary:**

The paper proposes a method to not only build a predictive model of human attributions based on robot motions as has been done in prior work, but to use this model to generate trajectories with those attributes. The generated trajectories are constrained to a good enough task performance. To this end, human attributions from a Mechanical Turk study are used to probabilistically map attributions to robot trajectories in a coverage task. Trajectories are then generated to optimize the Kullback-Leibler divergence of the behavior attribution model relative to a desired behavioral score factor distribution. This optimization is subject to an inequality constraint on the task cost. The value of the inequality is a design parameter that dictates how suboptimally the robot can behave in service of optimizing the behavioral attributes. The generated trajectories were then entered into a separate Mechanical turk experiment assessing how well the generated trajectories were perceived by humans. The approach was somewhat successful in eliciting the intended human perceptions by the participants.

**Issues:**

A revision should make the tables into Tables separate from any plots or graphs. The revisions should also clarify the reasoning for the low dimensional trajectory representation and provide more details on the optimal search for trajectories (at least in an appendix).

**Reviewer Expertise:**

Fair: Some knowledge of the area

**Strengths And Weaknesses:**

Strengths
This work goes beyond prior work for extracting human perception and attributions for robot motions and tries to integrate these models into the motion generation pipeline similar to generation of legible robot motion.

The experiments were well-designed. The authors were able to build a reasonable and understandable model of human attributions mapped to trajectories. The formulation of a cost for matching the probabilistic model could be valuable in other scenarios where one would generate trajectories with higher level behavioral attributes.

Weaknesses
A major weakness of the paper was the lack of detail in the optimization. The paper describes a hill-descending search in the space of trajectories, but why is this in the space of trajectories when section 4.1.2 says that the space of trajectories is `too large to explore directly’? It appears that the configuration space is already discretized into states. (This is implied by the comments on the state transition costs.) I think that what the paper means is that the space of trajectories is too large to map attributions to directly, since the described optimization is clearly exploring over \xi not \phi. I understand the naïve action level choices would be included, but more detail is needed on how to choose motion templates and encode them in the optimizations. In its current form, an average graduate student would be daunted by the task of replicating this or applying it to a robot arm without the relevant details.

I would also like to know why the factors are between -3 and 3. It seems like an arbitrary choice, so why this one in particular what are the consequences of making this domain larger or smaller?

Figure/Table Issues:
There are tables listed under Figure 2, 3, and 4. It seems bad practice to include tables as subfigures. They should just be tables using the \tabular environment. The paper also refers to fig. 2a as table 2a, which makes it inconsistent with itself. This practice also has made the text in fig./table 2a very small and hard to read. Fig/table 3a seems to have all the same data as what is contained in the plot in fig. 3b, so why not just drop the table altogether and leave the plot. Finally, figure 4 is hard to follow. I did not initially realize that it was two column all the way down until it hits the table which breaks the two column format above it.


**Summary Of Recommendation:**

The paper demonstrates a comprehensive approach to generating robot trajectories that exhibit behavioral attributes from collecting data, building a model, and exploiting that model to generate trajectories. The generation of trajectories is a significant advance over past work that only classified trajectories. However, some significant details about the optimization process is missing and the presentation of graphical and tabular data makes the paper somewhat difficult to digest.

Post Rebuttal Update: The author's have added significant clarifications and addressed most of the concerns of myself and the other reviewers. For this reason, I have updated my score to weak accept.

---

> ### Author Response · Authors · 2021-08-27
> **Authors' Response**
>
> Thank you for the feedback. We appreciate the work it takes to provide constructive comments.
>
> “lack of detail in the optimization”: We have added a new appendix that describes details of our trajectory optimization approach.
> * “why is this in the space of trajectories...”: The choice of space to search in is an important aspect of the optimization which we’ve made central to our description in the new appendix. To summarize, optimizing in the space of features is easier at first (since the model affords gradient descent), but solving the resulting inverse problem of mapping the optimized features back into a trajectory is difficult in practice because many combinations of features simply cannot be realized. It is more workable to begin with a feasible trajectory and transform the trajectory towards an increase in the objective function. The details of how to transform the trajectory bear explaining too, so we have described the transformations we used in the appendix as well.
> * “...section 4.1.2 says that the space of trajectories is `too large to explore directly…  I think that what the paper means is that the space of trajectories is too large to map attributions to directly’”: You are correct. Our intent with that statement was to express the difficulty of mapping the large space of trajectories down to attributions directly. We have changed this language in the paper.
>
>
> “why the factors are between -3 and 3”: The basis for this design wasn't documented in the paper. We’ve added an explanation. To summarize: Factor scores are standardized measures, meaning that they express how many standard deviations from the mean a response is. Capturing the range from -3 to 3 is expected to capture 99.7% of responses. There isn’t a particular requirement that someone applying our pipeline would have their attributions be factor scores (for instance, they could instead be framed as “how ____ is this on a 100 point scale” instead). We chose the question battery -> factor analysis -> factor scores approach because it is well documented and studied in the realm of psychometrics.
> * “What are the consequences of making this domain larger or smaller”: In our experiments, -3 to 3 is conservative; we did not observe responses that fell outside this range, so making the range larger would not impact the model. Making the range smaller would collapse the tails of the distribution into equivalence, progressively removing the model’s ability to distinguish between how extreme an attribution response is expected for a trajectory. An additional practical consequence of the [-3,3] domain is that it determines the scale of the NLL value for the uniform model. Shrinking this domain would increase the density predicted at each point, decreasing the loss. The scale of the NLL values for the learned models would also be expected to decrease at some rate. So it is important that the domain be the same if two models are to be compared using NLL.
>
> “bad practice to include tables as subfigures”: We’ve amended the paper to move tables out of subfigures.
>
> “Fig/table 3a seems to have all the same data as what is contained in the plot in fig. 3b”: the difference between these plots was not sufficiently explained in the text. The table reports the results that led us to select an ensemble model over alternatives. The secondary figure reports a separate sensitivity analysis (for which a different method of data folding was required, making the numbers not directly comparable) which let us know that we were seeing diminishing returns from collecting more data. We have moved this sensitivity analysis to an appendix as it is not central to the paper.
>
> “figure 4 is hard to follow. I did not initially realize that it was two column all the way down”: We have amended the paper to better integrate the discussion of the results with figure 4, providing more guidance on interpretation. Note that as a result of these changes, figure numbers have also changed. We have also added visual cues to the figure to indicate the two column structure.

---

### Official Review · Reviewer_ufvu · 2021-07-24

**Originality:** Very Good
**Technical Quality:** Very Good
**Clarity Of Presentation:** Very Good
**Impact:** 4

**Recommendation:**

Weak Accept: I recommend accepting the paper, but will not argue for my recommendation if the majority of other reviewers have a different opinion.

**Summary:**

This paper is studies the problem of inferring high-level behavioral attributes for robot motion plans. The work builds on a user study and psychology literature to to postulate a space of behaviors and features that are expected to inform behavioral attribution. Central to the approach is learning a multi-model distribution over attributes given a trajectory. The model is trained from crowd-sourced data and is used to synthesize trajectories that can balance motion cost and the expression of specified attributes. The model is validated using a user study.

**Issues:**

- While reading the paper, I was less clear on how the trajectory data set was created. I understand that the trajectories were generated by varying the parameters of a conventional motion planner and were later annotated via crowd sourcing.  I would request more technical details. Alternatively, human turkers could have drawn trajectories on the map for specific behavioral attributes to more directly elucidate their mental model.
- How extensible is the trajectory generation process to richer behavioral specification? For example, can the framework handle the case where a human specifies that a particular attribute “should" be present and another distinct attribute "should not” be present in the synthesized trajectory.


**Reviewer Expertise:**

Very good: Comprehensive knowledge of the area

**Strengths And Weaknesses:**

Strength
- The problem addressed in this work is well motivated and clearly articulated. Authors present a simple but elegant solution for the problem of incorporating behavioral attributions to robot motion planning.
- The experiments and user study is well thought out and rigorously conducted.

Weaknesses
- The problem is confined to metric characteristics of the robot’s trajectory and the overall coverage attained by the planner in the environment. Intuitively, it appears that symbolic properties of the goal and map context would also play a role in behavioral attribution. For example, a robot moving around a glass vase quickly may be considered rushed/careless but the same may not be interpreted if the robot moves around other furniture. Richer symbolic attributes of the goal and the environment lie beyond the present approach.
- The experimental evaluation does not explicitly compare against a baseline. Would recommend either comparing with an approach for related tasks such as legibility or state that the suitable baseline does not exist.

**Summary Of Recommendation:**

The paper clearly articulates the technical gap and presents an interesting approach to learning a model for behavior attribution and then using the model to synthesize plans that balance cost-to-goal with its behavioral attribution.

---

> ### Author Response · Authors · 2021-08-27
> **Authors' Response**
>
> Thank you for your feedback. We appreciate the constructive feedback and the effort spent considering the paper.
>
> “Richer symbolic attributes of the goal and the environment”: This is a good observation; discrete or symbolic attributes would not be well expressed in our framework. In some circumstances, there may be acceptable continuous analogs to a symbolic attribute. In the example of a robot rushing past a glass vase, it may be acceptable to express the feature as a percentage of some fixed amount of “time spent rushing past glass vases,” on the basis that the attribution response (anxiety) increases as the amount time spent rushing increases. However, indeed, incorporating symbolic attributes was beyond the scope of this paper and presents an exciting direction for future work.
>
> “comparing ... or state that the suitable baseline does not exist”: We have amended the paper to state that the lack of baselines for generating motion that balances high level attributions and task-directed motion.
>
>  “I was less clear on how the trajectory data set was created:” We have included new details in Appendix A describing how we bootstrapped the trajectory dataset. In summary:
> An author manually demonstrated six trajectories that they judged to maximally express different items (curious, broken, energetic, lazy, lost, scared)
> We collected responses to these trajectories and conducted an initial analysis of what aspects of the motion users said contributed to their ratings. This formed an initial subset of the features we would use.
> Additional trajectories were then generated using a hill-descending search in the space of trajectories optimizing for activating individual features while holding other features constant
> These new trajectories were posed to users for their ratings. Simultaneously, we asked them to provide demonstrations of how they would make the robot look “______” for random items from our questionnaire (much as you suggested).
> This feedback fed back into analysis and the process was cycled through two more times with a subset of the generated and demonstrated trajectories, resulting in the final dataset and trajectory featurization.
>
> “How extensible is the trajectory generation process to richer behavioral specification”: The limits on the form of the behavior specification come from the options we have to express a distance or divergence between the model’s predictions and the resulting factor scores. If we consider your example of a specification that “one attribution should be high and another should be low,” we can express this by placing a distribution centered at this point in the attribution space (say, [-1.5, 1.5]) and then express the difference between the model’s predictions and this goal as the KL divergence (so, just like our experiments, but now in two dimensions). Any extraneous dimensions of attribution that aren’t specified in the goal can be marginalized out to express indifference to the dimension.

---

### Official Review · Reviewer_AY1D · 2021-07-25

**Originality:** Very Good
**Technical Quality:** Excellent
**Clarity Of Presentation:** Good
**Impact:** 4

**Recommendation:**

Weak Accept: I recommend accepting the paper, but will not argue for my recommendation if the majority of other reviewers have a different opinion.

**Summary:**

- The authors formulate a computational model for a human's classification of the behaviour typified by a robot's trajectory
- In doing so, they are also able to formulate the inverse problem: a model to generate robot trajectories to best typify a desired behaviour
- In a simulation domain, the authors had AMT workers classify sets of robot trajectories. An exploratory factor analysis revealed three factors that the humans' evaluated the robot trajectories upon. These were the target behaviour types in a subsequent experiment
- The authors generated a model that was able to classify the likelihood of ratings of multiple people along the three factors (behaviours) for any given trajectory [as mentioned later, I am slightly confused on this point though]
- The authors use this model, and a hill-climbing search algorithm, to then generate trajectories in a new environment that try to satisfy a desired rating from humans.
- The authors find that they are able to predict participants' scores on the new trajectories at better than chance


**Issues:**

(See scope for improvement; above)

**Reviewer Expertise:**

Fair: Some knowledge of the area

**Strengths And Weaknesses:**

**Strengths**

- I like the computational formulation of the problem. However, I commend the authors even more on their data-driven approach to the issue of getting the behaviour description objectives (the exploratory factor analysis)
- Qualitatively, many of the results presented by the authors make sense:
  - The images of the trajectories (probably cherry-picked to an extent?) would conform to the adjectives that the authors ascribe to them
  - The lower predicted KL-divergence of generated broken trajectories, when the suboptimality bound is high, also makes sense
  - The distributions of 4b are a good sign of the authors having accomplished part of what they set out to do
- The results from the user studies are compelling evidence of the efficacy of this work.

**Scope for Improvement**

- The figures are hard to parse and it takes a while to know what the main result depicted in them are. 4c is particularly difficult. Focusing on that one, I think the authors are trying to showcase three results here:
  - The ability of the model to provide a prediction of how humans might rate trajectories
  - The ability of their framework to elicit human ratings that conform to a desired set of behaviour ratings
  - The agreement of their models' predictions of human ratings to the actual ratings

As such, the plots for experiment 1 then serve a different purpose than those for experiment 2. Therefore I recommend the authors separate those and particularly focus on purpose 2 (above) for the plots from experiment 2

- Related to the above point, for experiment 2, I think it's more important to showcase the divergences of participant ratings from the _target_ distribution of ratings instead of with the models' predictions. I suspect, given the current plots, that this might even lead to a more interesting story.
- The authors couch their results in the context of hypotheses: however, they do not provide the details of any corresponding hypothesis tests (either Frequentist with p-values or Bayesian with ROPE / pd). As such, it's hard to say whether the hypotheses are supported (or the corresponding null hypotheses are rejected). I recommend the authors remove such language unless they perform and show the results of _quantitative_ hypothesis tests.
- The text mentions a suboptimality condition of 12x; yet in the appendix, it is plotted as 2^3. Which is correct?


**Summary Of Recommendation:**

Although the paper has some weaknesses, they can be addressed by improving the presentation of the evaluations, as mentioned above. If this is done, I'm in favour of including this paper in the technical program.

Post-Rebuttal Update:
I thank the authors for their clarifications and find that the appendices, particularly Appendix D, is very helpful. I am much more in favour of accepting this paper into the technical program, but I am curious to see what my fellow reviewers say.

---

> ### Author Response · Authors · 2021-08-27
> **Authors' Response**
>
> Thank you for giving us constructive feedback. We know that this requires a substantial amount of time and effort.
>
> To what extent are images of trajectories cherry-picked: In a strict sense, they were not manually selected from a set of candidates so they weren’t cherry picked. We found that our optimization procedure reliably fell into the same solutions for each condition, meaning we had no opportunity to hand-select preferred alternatives to use in our experiments. We have amended the paper to note this.
>
> “Figures are hard to parse… 4c is particularly difficult”: We have made some visual modifications to better indicate the continuous two column structure and we have modified the discussion to better integrate the component figures, which should provide guidance on interpretation.
> * [Comments on the purpose of figure 4c (now labeled just figure 4)]: The intended purpose of this figure is to provide a qualitative impression for the correspondence of the model’s predictions and the empirical distributions (purpose 3 on your list). These graphs provide flavor and detail to the table below, which is a more precise quantitative characterization of how well the model predicted participant’s responses.
>   * We agree that purpose 2, “the ability of their framework to elicit human ratings that conform to a desired set of behaviour ratings,” is important to demonstrate. We express this as our third hypothesis, that modulating the suboptimality parameter lets the approach progressively increase the elicitation of an attribution. We have provided additional quantitative characterizations (correlation analyses) of the extent to which the experiments support this statement.
>
> “for experiment 2, I think it's more important to showcase the divergences of participant ratings from the target distribution of ratings instead of with the models' predictions.”: We agree, it’s important to characterize how close the generated trajectories are to the targeted specification. We didn’t illustrate this initially because the target distributions are the same in all experiments, a gaussian centered at 1.5. We have added a new dashed line representing this PDF to the plots. As the predicted distributions (grey line) show, in most conditions the generated trajectories aren’t predicted to come anywhere close to matching this target. Rather, they slowly and slightly move their mass towards the right. We characterized how the target divergence slowly decreases as the suboptimality bound is increased in Appendix D. This plot also depicts how even in the limit (as depicted by the bound line), the model does not believe that it can produce a trajectory that matches the goal distribution for any attribution in either environment.
>
> Hypothesis tests: We’ve amended the paper to report additional statistical results. In summary, we use bootstrapped confidence intervals to support the inference that the model remains superior to a null model (uniform baseline) in both of our experiments, and we conduct correlation tests to assess whether relaxing the suboptimality bound enables generating trajectories with stronger attributions, seeing mixed results depending on the attribution.
>
>
> “The text mentions a suboptimality condition of 12x; yet in the appendix, it is plotted as 2^3. Which is correct?”
> 12x was used as the fourth condition in the experiments reported in the main paper. The sensitivity analysis presented in the appendix reports the results of a separate experiment for which we generated trajectories smoothly varying between 1x and 16x suboptimality.

---

### Meta-Review · Area_Chair_Nwea · 2021-08-14

**Recommendation:** Accept (Poster)
**Confidence:** 4

**Metareview:**

This paper addresses the issue of inferring high-level behavioral attributes for robot motion planning, and presents a framework to involve human attributions from a Mechanical Turk study and then used to evalute and generate robot motion trajectories in a coverage task.

I agree with reviewers that the problem this paper addressed is well motivated and formulated, and the experiments is also well-designed, it's good to see the user study takes supports from psychology literatures.

While the presentation on both the details of the methods and analysis of expreimental results seems a bit weak, as reviewers pointed out. I agree with these points which might be valuable in improving the quality of the paper, mainly including:

1. The lack of details in the optimization process.
2. The experimental evaluation and analysis needs to be greatly improved, like explicitly compare against a baseline, carefully contrast the model with human, the results of quantitative hypothesis tests, etc.
3. Many more minor points as reviewers mentioned.

More added:
I would like to thank authors for your carefully and detailed responses to comments raised by reviewers.
Although there are still rooms for this research to be improved, the updated version seems better, and the scores of this paper was improved after rebuttle.
I thus recommend this paper to be accepted.

---

> ### Author Response · Authors · 2021-08-27
> **Authors' Response**
>
> Thanks for the constructive feedback and the specific pointers that helped us greatly improve our paper!
>
> We would like to highlight that all reviewers found our evaluation and analysis of our experimental results to be strong. Specifically, AY1D found our evaluation to present “compelling evidence of the efficacy of our approach” whereas ufvu notes that our “experiments and user study is well thought out and rigorously conducted” and zJU5 agrees stating that our experiments “were well-designed”.
>
> Further, AY1D commends our computational framework and our data-driven approach for getting the behaviour description objectives. ufvu commends the simplicity and elegance of our approach and zJU5 finds that “the generation of trajectories is a significant advance over past work that only classified trajectories”. Finally, all reviewers found our work to be well-motivated and clearly described, and appreciated its originality and potential impact.
>
> We have incorporated the feedback provided by all reviewers (see below for specific comments and pointers to all reviewers comments). To address the main points brought up by the metareviewer:
>
> 1. We have added a new section, Appendix D, discussing our optimization framework in more depth. This section describes all steps involved in optimizing a trajectory to elicit a desired attribution.
> 2. We have expanded the discussion on the evaluation of our model by addressing the three main areas pointed out by the meta reviewer:
>
>     * Baseline: To our knowledge, there are no suitable baselines because no prior work has proposed methods for eliciting high level attributions while balancing task performance. As mentioned by zJU5, “the generation of trajectories is a significant advance over past work that only classified trajectories”. We have amended the paper to include this explanation.
>     * “Contrast the model with a human”: Direct comparisons of the approach’s ability to generate trajectories that elicit desired attributions with the abilities of human experts to do the same are an interesting direction for future evaluation. Such evaluations raise new fairness difficulties, like the selection of the human experts as well as the details of their training and instructions. We have added a note highlighting this idea in our discussion.
>     * Quantitative hypothesis tests: We’ve added statistical results.
> 3. We hope we have been able to clarify and amend the paper to address the points raised by the reviewers. Many of these changes centered on the presentation and interpretation of figures and tables, which we have rearranged, clarified and better integrated into the paper.

---

### Decision · Program_Chairs · 2021-09-13

**Decision:**

Accept (Poster)

**Comment:**

This paper addresses the issue of inferring high-level behavioral attributes for robot motion planning, and presents a framework to involve human attributions from a Mechanical Turk study and then used to evalute and generate robot motion trajectories in a coverage task.

I agree with reviewers that the problem this paper addressed is well motivated and formulated, and the experiments is also well-designed, it's good to see the user study takes supports from psychology literatures.

While the presentation on both the details of the methods and analysis of expreimental results seems a bit weak, as reviewers pointed out. I agree with these points which might be valuable in improving the quality of the paper, mainly including:

1. The lack of details in the optimization process.
2. The experimental evaluation and analysis needs to be greatly improved, like explicitly compare against a baseline, carefully contrast the model with human, the results of quantitative hypothesis tests, etc.
3. Many more minor points as reviewers mentioned.

More added:
I would like to thank authors for your carefully and detailed responses to comments raised by reviewers.
Although there are still rooms for this research to be improved, the updated version seems better, and the scores of this paper was improved after rebuttle.
I thus recommend this paper to be accepted.